# Direct X-ray photoconversion in flexible organic thin film devices operated below 1 V

Laura Basiricò[1], Andrea Ciavatti[1], Tobias Cramer[1], Piero Cosseddu[2], Annalisa Bonfiglio[2] & Beatrice Fraboni[1]

The application of organic electronic materials for the detection of ionizing radiations is very appealing thanks to their mechanical flexibility, low-cost and simple processing in comparison to their inorganic counterpart. In this work we investigate the direct X-ray photoconversion process in organic thin film photoconductors. The devices are realized by drop casting solution-processed bis-(triisopropylsilylethynyl)pentacene (TIPS-pentacene) onto flexible plastic substrates patterned with metal electrodes; they exhibit a strong sensitivity to X-rays despite the low X-ray photon absorption typical of low-Z organic materials. We propose a model, based on the accumulation of photogenerated charges and photoconductive gain, able to describe the magnitude as well as the dynamics of the X-ray-induced photocurrent. This finding allows us to fabricate and test a flexible $2 \times 2$ pixelated X-ray detector operating at 0.2 V, with gain and sensitivity up to $4.7 \times 10^4$ and 77,000 nC mGy$^{-1}$ cm$^{-3}$, respectively.

[1] Department of Physics and Astronomy, University of Bologna, Viale Berti Pichat 6/2, Bologna 40127, Italy. [2] Department of Electrical and Electronic Engineering, University of Cagliari, Piazza D'Armi, Cagliari 09123, Italy. Correspondence and requests for materials should be addressed to L.B. (email: laura.basirico2@unibo.it) or to B.F. (email: beatrice.fraboni@unibo.it).

The research interest on alternative materials for innovative ionizing radiation detection is rapidly growing. Traditional materials for direct, solid-state detectors (that is, directly converting ionizing radiation into an electrical signal) such as silicon, cadmium zinc telluride or mercury iodide ($HgI_2$), suffer from severe limitations in their processing into large-area pixelated detector matrices[1]. A new generation of ionizing radiation sensors has to be envisaged, ideally combining ease of processing, low power supply and mechanical flexibility. Indeed, several applications, spanning from citizens' security, to industrial and to medical diagnostics, require thin, conformable sensor panels, for a large-area determination of the incoming radiation dose and energy distribution. As such applications are of high commercial interest, intense efforts have been recently devoted to the realization of large-area direct detectors based on inorganic materials (for example, amorphous Silicon[2], amorphous Selenium[3], Diamond[4]), which, however, still share the constraints of expensive or complex growth techniques and maintain stiff mechanical properties. An alternative approach is based on organic materials as active photodetectors. One of the appealing features of organic materials is the easy deposition over large areas by means of solution-coating, low-cost techniques, such as drop casting, inkjet printing[5,6], solution shearing[7], potentially onto flexible substrates[8]. Moreover, the similarity of the typical density of organic molecules to that of human tissue makes them very interesting for medical X-ray direct dosimetry[9]. In the field of ionizing radiation detection, there are two possible approaches: indirect radiation detection systems, where the radiation is at first absorbed by a scintillating material and then converted into an electrical signal by a photodetector, and direct detectors, in which the photons are directly converted into an electrical signal. Organic molecules and polymers have been employed in indirect devices as scintillators or as active layer of the photodetector[10–12]. Very recent reports show how hybrid combinations of inorganic/organic scintillator–photodetector systems can provide thin, large-area ionizing radiation detectors with very good performances[12,13]. However, none of these innovative systems is flexible nor works with ultra-low voltage bias.

In the last years, a few works reported the proof-of-principle for direct X-ray detection based either on organic semi-conducting single crystals[14–17] or on polymer thin-films blended either with π-conjugated small molecules[18], inorganic high-Z nanocomponents[19,20] or carbon nanotubes[21] to enhance the sensitivity to X-rays improving the charge carriers mobility and the stopping power of the material.

In this work, we investigate direct X-ray photo-conversion in micro-crystalline thin films of bis-(triisopropylsilylethynyl) pentacene (TIPS-pentacene) (a standard material for the fabrication of organic devices onto flexible plastic substrates[22]) deposited by drop casting onto flexible poly(ethyleneterephthalate) (PET) substrates. We find that these devices are characterized by an unexpected high X-ray sensitivity that we justify by interpreting the detection mechanism as a photo-modulation of the semiconductor conductivity due to charge accumulation during X-ray exposure, resulting in a photoconductive gain effect. From such findings, we develop a kinetic model that gives an important insight into the physical process that leads to highly sensitive response to ionizing radiation by such low-Z organic materials. Furthermore, this investigation allowed us to realize a flexible, pixelated X-ray detector based on organic thin films, operating at very low voltage (below 1 V). The ultra-low voltage operation paves the way towards battery-operated wearable detectors.

## Results

**X-ray photocurrent in TIPS-pentacene organic thin films.** The detectors consist of a 100 nm thick TIPS-pentacene organic active layer deposited by drop-casting onto interdigitated gold electrodes fabricated on 125 µm thick PET film. Figure 1a,b depict a schematic of the structure and a photograph showing the flexibility of the device, respectively. The bottom-up deposition process, based on drop-casting of a solution containing TIPS-pentacene, gives rise to high-quality micro-crystalline films with crystallites extending a few tens of micrometres in length and width as measured by atomic force microscopy (AFM, Fig. 1c). The average crystallite height is 100 nm. Figure 1d shows the typical X-ray photocurrent signal of a device biased at 0.2 V. While recording the current, three on/off switching cycles of a monochromatic synchrotron X-ray beam of 17 keV (dose rate 19.3 mGy s$^{-1}$) are performed. The energy of radiation is chosen to be in line with the typical values of medical diagnostic analysis (for example, mammography), assessing therefore the performance of the device in medically relevant conditions. About 20 devices have been characterized with no significant differences in the detection performance (statistical variation of 15%, see Supplementary Fig. 1). The X-ray-induced photocurrent takes about 60 s to saturate at a value of 2.8 nA. After switching the X-ray photoexcitation off, the current takes a comparable amount of time to return to its baseline value. The contribution of the electrodes and substrate to the X-ray-induced photocurrent can be evaluated by comparing the signal amplitude of detectors with metal electrodes with that of fully organic devices (see Supplementary Fig. 2). The photocurrent is also recorded at different bias voltages and the plot of the signal amplitude versus bias voltage is reported in Fig. 1e. The X-ray-induced photocurrent increases almost linearly with the applied voltage. However, we set the operation voltage in the low voltage range (0.2 V) to avoid possible instability issues due to bias stress effect (see Supplementary Fig. 3) and to avoid the decrease of the detector signal-to-noise ratio (that is, the ratio between the X-ray-induced photocurrent and the current in dark) due to the observed superlinear increase in the dark current due to the onset of space charge effect (see Supplementary Fig. 4). The observed magnitude of the photocurrent produced by high-energy photons in a thin organic layer rules out an explanation purely based on the collection of photo-generated charge carriers, usually applied to inorganic semiconductors[23], as explained in the following. Below we propose a model that describes the modulation of the conductivity of the organic thin-film by the accumulation of charge carriers in the material, giving rise to a photoconductive gain[23–25] and allowing the explanation of the experimentally measured magnitude of the signal as well as its dynamics.

**Charge collection in the X-ray response of organic detectors.** Generally, the absorption of an X-ray photon in an inorganic semiconductor results in the creation of a high-energy primary electron, which deposits its excess kinetic energy in the surrounding medium in the form of electron-hole pairs and phonons[23]. Under the effect of an applied electric field, the electron-hole pairs separate and drift to the respective electrodes, thus generating a photocurrent. Recombination and phonon formation represent losses. The magnitude of the photocurrent $I_{CC}$, accounting for such a carrier collection process, is described by $I_{CC} = \Phi n q$, where $q$ is the elementary charge, $\Phi$ is the photon absorption rate, $n$ is the number of the generated electron-hole pairs per absorbed photon[23]. To apply the above considerations to interpret and model the current experimentally recorded on the here investigated organic thin films, we introduce

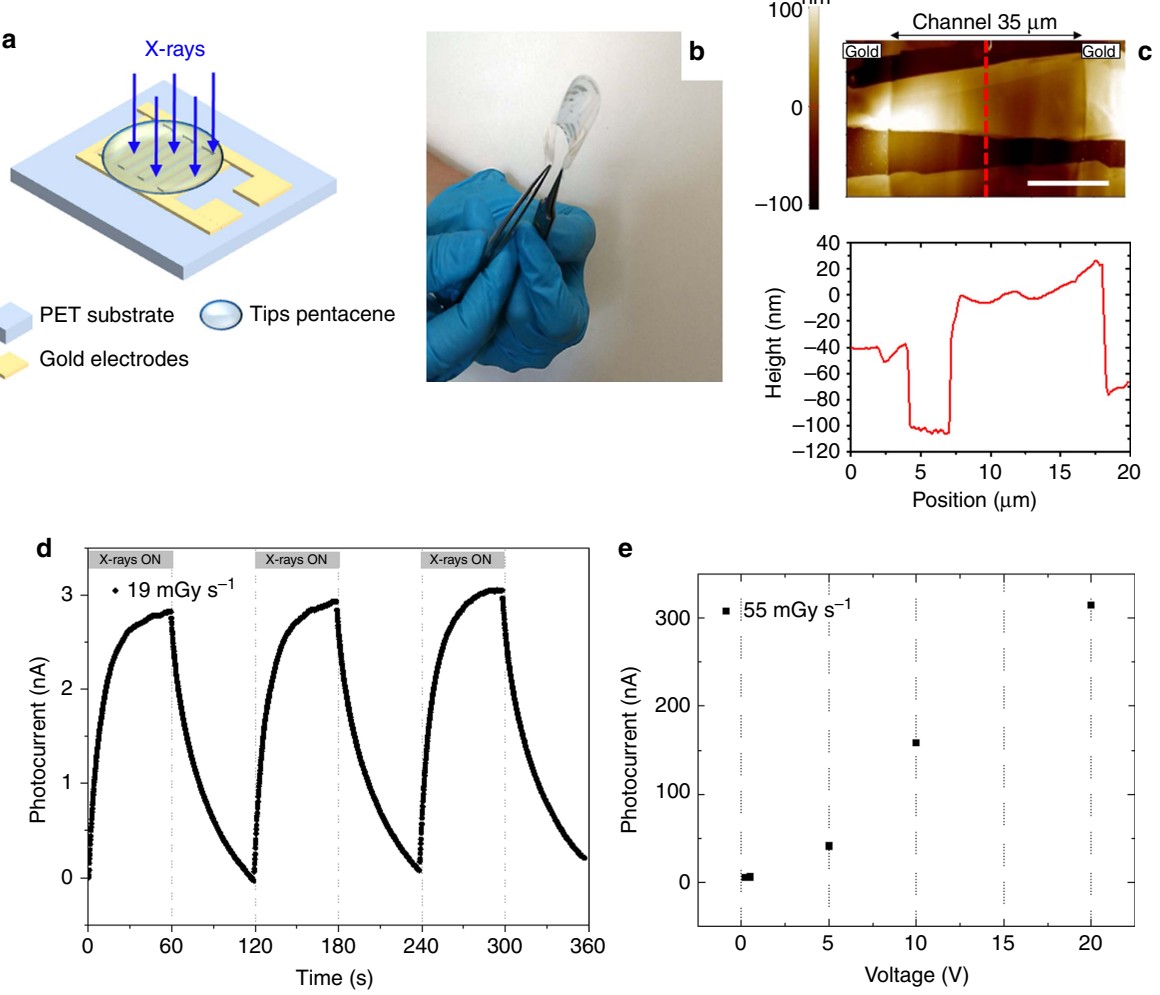

**Figure 1 | Flexible organic X-ray photodetectors based on TIPS-pentacene thin films. (a)** Schematic of the device structure. The organic active layer is drop-casted onto gold electrodes pre-deposited on flexible PET substrate in an interdigitated configuration. **(b)** Image showing the flexibility of the structure. **(c)** AFM micrograph (top) and height profile (bottom) of the TIPS-pentacene crystal domains within the channel region (white scale bar: 10 μm). **(d)** X-ray-induced photocurrent signal recorded at a bias voltage of 0.2 V, upon three on/off switching cycles of a monochromatic synchrotron X-ray beam at 17 keV. **(e)** Photocurrent signal at different bias voltages recorded under an X-ray beam by a Mo-target X-ray tube with a dose rate of 55 mGy s$^{-1}$.

the geometry of the semiconducting layer as $A = 0.015$ cm$^2$ (active area) and $h = 100$ nm (thickness). Further, to obtain an upper limit estimation, we consider $n = E_{ph}/E_g$, where $E_{ph}$ is the photon energy, and $E_g \approx 2$ eV is roughly the energy gap of TIPS-pentacene. Due to the low photon absorption of the organic semiconductor (attenuation length = 0.68 cm; absorption = 0.0015%, see Supplementary Note 1), we obtain photocurrent values of $I_{CC,max} < 2$ pA. Despite this estimate represents an upper limit, the value is about two orders of magnitude smaller than the observed signal amplitude.

Consequently, other processes must be involved in the generation of such a large photocurrent, that we ascribe to an increase in conductivity due to a photoconductive gain, arising when X-ray generated, free charge carriers accumulate in the organic semiconductor and pass several times through the organic material between the electrodes before recombination sets in. This mechanism leads to an amplified photocurrent:

$$\Delta I_{PG} = G\, I_{CC} \qquad (1)$$

with $G$ being defined as the photoconductive gain[25]. We exclude an X-ray-related hole mobility modification as hole mobility values calculated from transfer characteristics of TIPS-pentacene-based organic thin film transistors (OTFTs) exhibit no significant

differences if measured in the dark or under X-ray irradiation (Supplementary Fig. 5). Therefore, we propose that, during X-ray irradiation, additional free carriers are generated and accumulate in the organic thin film. Since the devices are made by gold electrodes forming ohmic contacts with TIPS-pentacene (as the work function of gold: $\phi_{Au} = (4.7 \div 5.2)$ eV[26] is generally considered matching to HOMO level of TIPS-pentacene: $\phi_{TIPS} = 5.3$ eV[27]), an increase in carrier concentration $\rho_X$ results in an increase of current $\Delta I_{PG}$ according to:

$$\Delta I_{PG} = Wh\rho_X\mu E \qquad (2)$$

with $E = V/L$ being the electric field and $W$ being the active width of the interdigitated structure[25].

To derive a model for the increase in free carriers and its impact on photocurrent we consider the differences in hole and electron carrier transport in organic materials[28]. A non-equally efficient transport can be due either to a difference in the charge carrier mobility between holes and electrons or to the presence of traps for one of the two kinds of charge carriers. It is well known that hole mobility in TIPS-pentacene can reach very high values[7]. On the other hand, electrons are known to get trapped very easily due to the employment of a polar substrate such as PET[29], and to the presence of oxygen (a well-known trap for electrons[6,30]), as it

is the case of our measurements that are carried out in ambient conditions. Here therefore, similarly to what has been proposed to interpret UV-vis induced photoconductivity in organic semiconductors[28,31], we assume that the additional electrons and holes generated by the interaction with the X-ray follow a different fate: holes drift along the electric field until they reach the collecting electrode while electrons remain trapped and act as 'doping centres'. To guarantee charge neutrality, mobile holes that are collected at the collecting electrode are continuously re-injected from the injecting electrode. Consequently, for each electron-hole pair created, more than one hole contributes to the photocurrent, resulting in a sort of 'doping' process and leading ultimately to a photoconductive gain. A schematic describing the process explained above is depicted in Fig. 2a. Crucial for the amplification in this mechanism is the slow recombination dynamics of X-ray generated carriers, resulting here from the presence of deep trap levels which remove free electron carriers from the recombination process.

To assess the proposed model based on carrier accumulation, we perform two investigations: first, we measure the variation in material conductivity by 4-probe experiments; second, we model the kinetics of the photo-response and derive the consequences for organic thin film X-ray detectors.

**X-ray-induced conductivity modulation.** Figure 2b reports the conductance as a function of time of the organic material upon three on/off X-ray beam cycles. The X-ray-induced conductance modulation is recorded employing both 4-probe and 2-probe electrical measurements on dedicated samples with co-linear electrodes configuration as shown in Fig. 2c. In the 4-probe measurement a constant current of 10 nA is flown through the outer electrodes, and the voltage drop between the inner probes is measured while switching on and off the X-ray beam. In the 2-probe measurement, on the other hand, the current is flown through the inner electrodes, where the voltage drop is also measured. The voltage drop occurring upon switching the X-rays beam on leads to a relative variation of the conductance (calculated as the ratio of the conductance variation after 60 s of irradiation and the conductance recorded before switching on the X-rays) of about 40% for both the 4- and 2-probe measurements. We infer from these observations that the X-ray exposure directly affects the bulk conductivity of TIPS-pentacene. The difference in conductivity observed with 4- and 2-probe measurements, due to contact resistance between the gold electrode and the TIPS-pentacene layer, can be ascribed to morphological non-homogeneities at the electrodes or to the formation of surface dipoles at the interface[32,33].

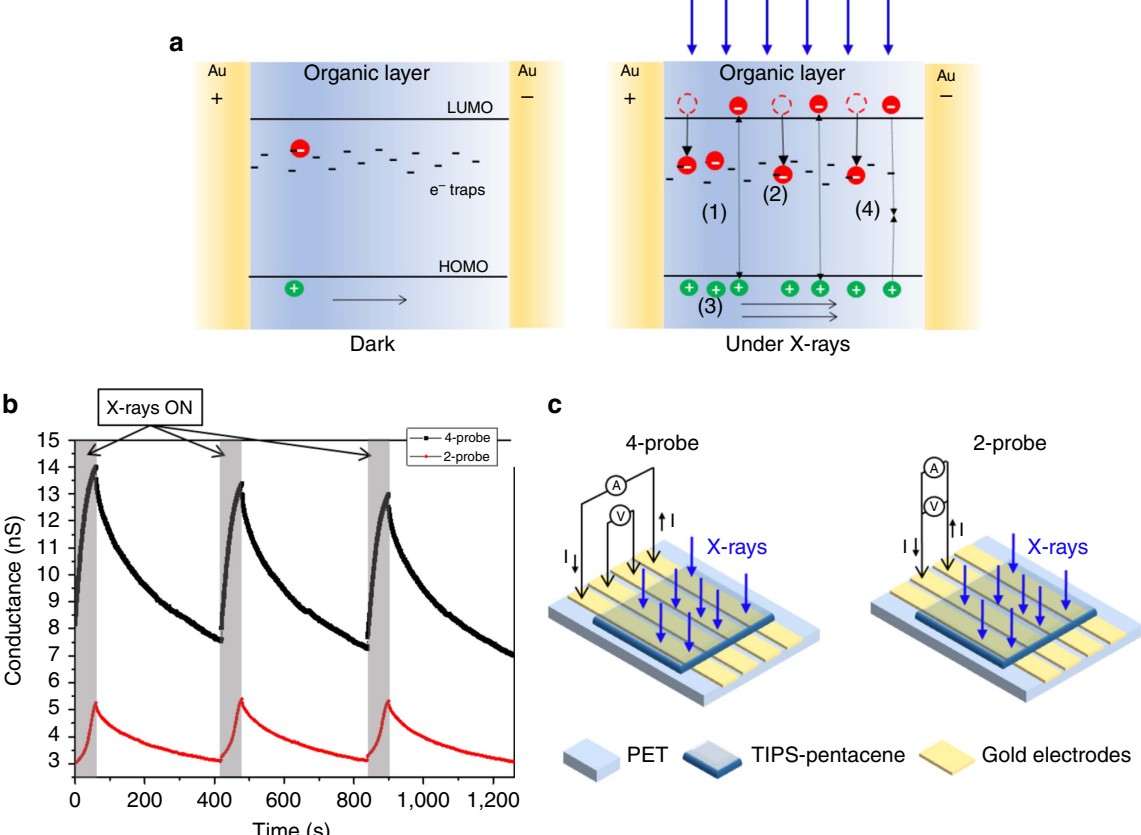

**Figure 2 | Charge accumulation in the X-ray photo-response of organic detectors.** (**a**) Schematic of the process of modulation of the conductivity induced by X-rays exposure of TIPS-pentacene thin films: (left) in dark the conductivity is due to the intrinsic carriers; (right) under X-ray irradiation: (1) additional electrons and holes are generated; after generation holes drift along the electric field until they reach the collecting electrode while (2) electrons remain trapped in deep trap states within the organic material. (3) To guarantee charge neutrality, holes are continuously emitted from the injecting electrode. As a consequence, for each electron-hole pair created, more than one hole contributes to the photocurrent leading to a photoconductive gain effect. (4) Recombination process takes place, counterbalancing the charge photogeneration in the steady-state. (**b**) Plots of the 4-probe (black squares) and 2-probe (red circles) measurements upon three switching on and off the X-ray beam (the grey regions in the plot indicate when the device is under irradiation). The X-ray source is a Mo-target tube with 35 kV of accelerating voltage. (**c**) Schematic of the electrode configuration used for the 4-probe (left) and the 2-probe (right) measurements reported in **b**.

**Kinetic model of the X-ray photo-response.** To model the experimental saw-tooth shape of the X-ray photocurrent induced by an on/off switching X-ray beam (Fig. 1d), we consider the variation of photo-generated carrier concentration $\rho_X$ in time:

$$\frac{\partial \rho_X(t)}{\partial t} = \frac{\Phi n q}{Ah} - \frac{\rho_X(t)}{\tau_r(\rho_X)} \qquad (3)$$

The concentration of intrinsic carriers $\rho_i$, created either thermally or by impurities, is assumed to give rise to a constant background current and is not further analysed. Equation (3) consists of two terms: the first describes the accumulation of carriers; here the only unknown parameter for the organic material is $n$, the amount of generated charges per absorbed photon. The second term describes the recombination of carriers and contains the free carrier lifetime $\tau_r(\rho_X)$. We approximate $\tau_r(\rho_X)$ by the phenomenological equation:

$$\tau_r(\rho_X) = \frac{\alpha}{\gamma}\left[\alpha \ln\left(\frac{\rho_0}{\rho_X}\right)\right]^{\frac{1-\gamma}{\gamma}} \qquad (4)$$

with $\alpha$, $\gamma$ and $\rho_0$ being material-specific constants describing the characteristic time-scale and the dispersion of trap states, and a reference carrier density, respectively (see Supplementary Note 2). Equation (4) gives rise to a stretched exponential recovery after exposure (see below) as experimentally observed. The combined equations (2)–(4) fully describe the dynamics of carrier photogeneration and recombination as well as the emerging photocurrent during and after X-ray exposure. To determine the parameters of the equations, we perform a numerical fit to reproduce the experimental photocurrent transients. In the calculations we set $\mu = 0.04\,\mathrm{cm^2\,V^{-1}\,s^{-1}}$ for hole transport as determined from field effect transistor measurements in structures similar to those prepared for the X-ray tests, but containing additionally a third gate electrode (see Supplementary Fig. 5). Figure 3a shows the result for three X-ray photocurrent signals obtained at different dose rates. The model well reproduces the saw-tooth shape of all experimental curves, using a single set of fitting parameters. Performing these fits for several photocurrent transients obtained from different devices we obtain: $n = 1,400 \pm 300$, $\alpha = (8.0 \pm 1.3)$ s, $\gamma = 0.60 \pm 0.07$ and $\rho_0 = (3.7 \pm 0.4) \times 10^{-5}\,\mathrm{C\,cm^{-3}}$. Saturation of the photocurrent at exposure times exceeding 60 s arises from the counterbalancing of charge generation and recombination processes in a steady-state dynamics. For the case of $D = 0$ (null radiation dose rate impinging on the material) the solution to equation (3) simplifies to a stretched exponential: $\rho_X = \rho_0 e^{-t^\gamma/\alpha}$ which describes the photocurrent decay due to charge recombination after switching off the X-ray beam. It has been demonstrated that a stretched exponential results when, instead of a single discrete trap level, a distribution of states is introduced into the kinetics, with $\gamma$ being the parameter which describes the broadness of the distribution[34] (see Supplementary Note 2 and Supplementary Fig. 6). Such a recombination kinetics has been observed earlier in UV-vis photocurrent decay[34] in amorphous[35,36] and organic semiconductors[37].

The proposed kinetic model supports carrier accumulation as the origin of the observed X-ray photocurrent: in TIPS-pentacene detectors, we identify the distributed trap levels as electron traps positioned deep in the band-gap. These trap-states remove excited electron carriers from the recombination dynamics and capture them for an effective time-scale $\tau_r$. The same time-scale then determines the lifetime of the mobile holes leading to the observed photocurrent. Deep electron trap states have been reported for several organic semiconductors at energies exceeding 0.8 eV below the conduction band[38]. Due to the presence of such trap sites, electron transport is suppressed and only effective hole transport is observed in many organic semiconductors. Following the definition of the photoconductive gain $G = \tau_r(\rho_X)/\tau_t$ with the transit time $\tau_t = L^2/\mu V$, we obtain values from $G = 2.6 \times 10^4$ (for 19.3 mGy s$^{-1}$ exposure) to $G = 4.7 \times 10^4$ (for 10 mGy s$^{-1}$ exposure) (see Supplementary Note 3 for detail). Similar values are reported for UV-vis organic photoconductors[39].

This model further permits to describe the dose rate dependence of the photocurrent, which determines the detector sensitivity defined as $S = \partial I/\partial D$. Figure 3b presents the related experimental data in the form of a plot of photocurrent as a function of dose rate, for various X-ray exposure times ranging from 5 to 60 s (the dynamic curve of response to the radiation for exposure time of 5 s is reported in Supplementary Fig. 7). All curves show a monotone increase with increasing dose rate and the maximum photocurrent is reached for the longest exposure times. However, with increasing exposure time, a transition from a linear dependence to a sublinear one is observed. The detector sensitivities $S$ as a function of dose rate are obtained from the curves derivatives and are plotted in Fig. 3c. The highest experimental sensitivity amounts to 180 nC Gy$^{-1}$ (72,000 nC mGy$^{-1}$ cm$^{-3}$) and is obtained for long exposure times and low dose rates. The model is in excellent agreement with these findings.

In fact, following equation (3) (see Supplementary Note 4 for detail) in the limiting case of short exposure time ($t \ll \tau_r$ blue line in Fig. 3b,c) the current response versus dose rate is linear: $\Delta I_{PG} = (t/\tau_t)I_{CC} = (t/\tau_t)S_{CC}D$ with $S_{CC}$ corresponding to the sensitivity of the equivalent detector purely based on charge collection. Therefore, as long as the exposure times exceed $\tau_t$ (here $\sim$1 ms), a gain effect is achieved in the detector. On the other hand, for longer exposure times ($t > \tau_r$), the photocurrent saturates and equation (3) simplifies due to steady-state dynamics ($\partial \rho_X/\partial t = 0$). The current response becomes $\Delta I_{PG} = \frac{(\tau_r(\rho_X))}{\tau_t}S_{CC}D = GI_{CC}$, that is no longer a linear relation since the carrier lifetime decreases with increasing charge density. By introducing equation (3), an implicit relation between $D$ and $\Delta I_{PG}$ is obtained which is plotted in Fig. 3b,c (black line; see Supplementary Note 4 for further detail).

We note that the non-linearity arises from the broad energetic distribution of the electron trap states. For the ideal case of a single discrete trap level ($\gamma = 1$), equations (3) and (4) simplify again to a linear response. We conclude that in organic X-ray detectors with amplified response due to photoconductive gain, the radiation exposure time $t$ becomes a crucial parameter which determines the sensitivity of the device: at long exposure times, very high sensitivities to the radiation are achieved, showing a maximum at low dose rates. Short exposition times lead instead to lower and constant sensitivities. In the first operation regime (long exposure time and low dose), maximum sensitivity values of up to 200 nC Gy$^{-1}$ (corresponding to 77,000 nC mGy$^{-1}$ cm$^{-3}$) for devices biased at 0.2 V have been found. These values exceed by more than two orders of magnitude those reported in the literature for organic devices based on thin polymeric films[19,20,40] or bulk organic single crystals[17], biased at several tens of volts. To the best of our knowledge, no other low-voltage organic X-ray detectors have been reported in literature so far.

**Flexible X-ray detector and realization of a detector matrix.** To evaluate the mechanical flexibility of the system, we characterize the detectors in a bent configuration. The bending radius is set to 0.3 cm, a value chosen to be conformable to most of the human body curves in view of possible medical diagnostic applications, by means of the experimental setup shown in Fig. 4a: the edges of the PET substrate are fixed into the clamps of the machine; when one clamp is moved towards the other, the substrate is bent. Figure 4b reports the X-ray induced

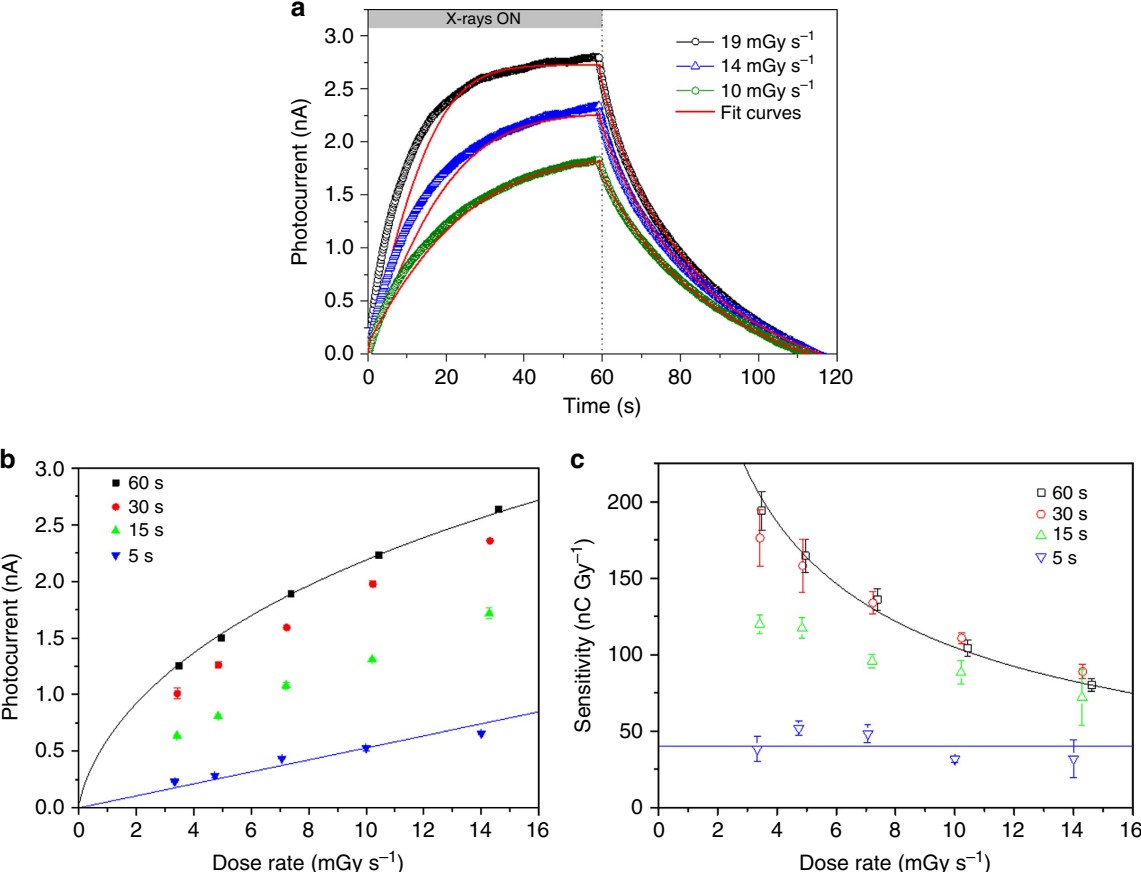

**Figure 3 | Dynamics of X-ray response and consequences on detector operation. (a)** Experimental and simulated curves of the dynamic response of the detector for three different dose rates of the radiation. The experimental data refer to 60 s of exposure of the device ($W = 48$ mm, $L = 30$ µm, bias 0.2 V) to a synchrotron 17 keV X-ray beam, with a bias of 0.2 V. **(b)** Photocurrent versus Dose rate plot (scattered points) recorded for different exposure times of the device to a synchrotron monochromatic X-ray beam at 17 keV and biasing the device at 0.2 V. **(c)** Sensitivity values, founded differentiating the photocurrent data in function of dose rate (reported in **a**), versus dose rate for different time exposures of the device to the radiation. Solid lines in **(a–c)** represent fits to the data according to the analytical model described in the text. The error bars in **c** are calculated by means of error propagation theory from errors of experimental data reported in **b**.

photocurrent for different dose rates of a device measured in three different experimental conditions: (1) before bending (black squares), (2) during bending (red circles) and (3) after bending (green triangles), with the substrate restored to a flat conformation. The measurement performed during the first bending (1 h long for a complete detector characterization for sensitivity determination) evidenced a decrease of about 50% of the X-ray-induced photocurrent. No significant differences (average signal variation below 5%) are noticed between following repetitive measurements on bent and relaxed devices. This behaviour suggests that the first bending leads to an irreversible degradation of the detector, which however keeps its response stable after that. Therefore, with the aim of reaching the electromechanical stability, the device is pre-bent before performing further mechanical tests described in the following. The demonstration of the mechanical robustness of the detector is reported in Fig. 4c: it shows the value of the maximum photocurrent measured under a 55 mGy s⁻¹ dose rate X-ray beam during bending and as a function of the number of bending cycles applied to the detector. In this plot error bars refer to the statistical fluctuations of the signal amplitude over three on/off switching cycles (60 s for each state) of the X-ray beam in the same condition. The average variation of the signal amplitude is below 8%.

Extending the here described organic thin film detector into a flexible detector matrix is easily achieved due to the simple and scalable processing steps employed in the fabrication of such

devices. We realize, as an example, a 2 × 2 pixel matrix (pixel size 5 mm) and test it under a monochromatic synchrotron X-ray beam at 17 keV. Such an energy value is chosen to be comparable to those commonly employed in medical diagnostic applications (for example, mammography) and therefore to test the device performance in medically relevant conditions. The electrical output signals from each pixel are simultaneously measured via a 4-channel picoammeter while a bias of 0.2 V is applied. Figure 5 shows the dynamic photocurrent responses recorded when: (a) only pixels 1 and 4 are irradiated (Fig. 5a, (b), only pixels 2 and 3 are irradiated (Fig. 5b) and (c) all the four pixels are irradiated (Fig. 5c). Note that the four signals of the pixels reported in Fig. 5a–c refer to the X-ray-induced photocurrent, that is, the dark current is subtracted, and the baseline of each plot is shifted in y axis for clarity. The schematic and the pixels indexing used are reported on the right of each graph, with a red box indicating the pixels under irradiation. All four pixels respond to the radiation with the same dynamic behaviour and the irradiated pixels can be clearly identified and easily distinguished from the non-irradiated ones. Moreover, as shown in Fig. 5c, the pixels exhibit comparable outputs when all irradiated within the same active area. To test the applicability of the device, a single pixel is used to perform an X-rays x-y scan of an aluminium annulus (optical image reported in Fig. 5d). The X-ray image, reported in Fig. 5e, shows high contrast, and it has a spatial resolution limited

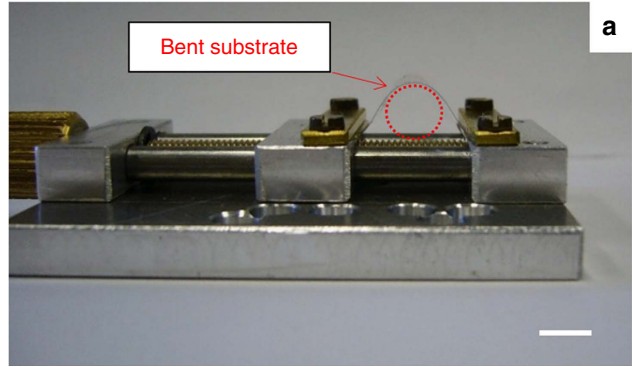

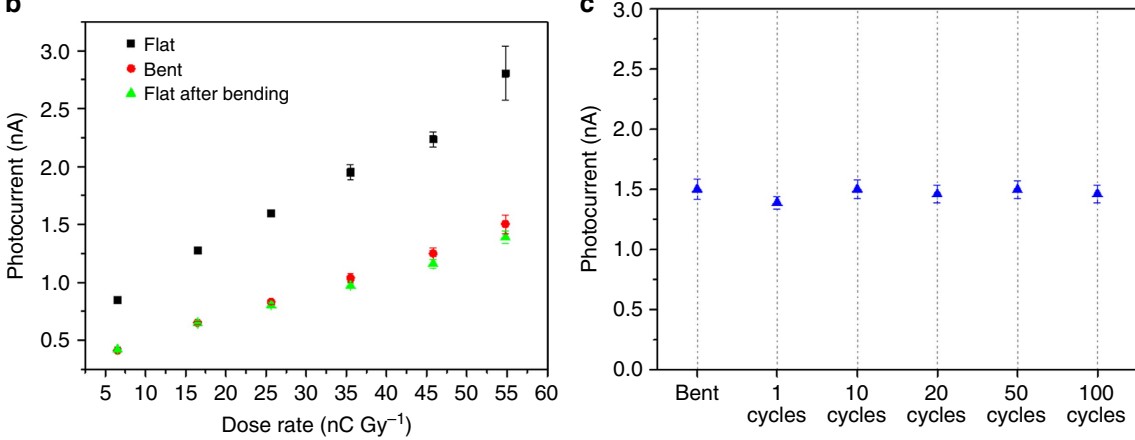

**Figure 4 | Assessment of the mechanical reliability of the system.** (**a**) Experimental set-up used for the characterization of the detector during bending (scale bar: 1 cm). (**b**) Photocurrent as a function of dose rate of a device measured before bending (black solid squares), during bending with a bending radius of 0.3 cm (red solid circles) and after bending (green solid triangles). (**c**) Photocurrent of a device measured under bending and in flat-substrate condition after the 1, 10, 20, 50 and 100 bending cycles, at the same bending radius $R = 0.3$ cm. The device ($W = 26$ mm; $L = 30$ μm), biased at 0.5 V, is irradiated under an X-ray beam provided by a Mo-tube with 35 kV of accelerating voltage and a dose rate of 55 mGy s$^{-1}$.

only by the detector dimensions and the X-ray beam collimation. The high reproducibility and reliability of the detector allow thus to achieve a good quality of the X-ray image, acquired by means of 750 X-ray exposures.

## Discussion

This work reports on novel flexible direct X-rays detectors based on organic semiconducting thin films, operating at very low bias voltage (below 1 V). The active layer of the detector is a microcrystalline film of TIPS-pentacene, deposited from solution onto flexible plastic substrate. Although the detector absorbs only a small fraction of the incoming high energy radiation (0.0015%), a strong photocurrent is observed together with highly sensitive direct X-ray detection. In order to describe and explain the amplitude and the dynamics of the photocurrent signal, we propose a model based on the accumulation of holes in the organic thin film during X-ray exposure, causing an increase in the material conductance and photoconductive gain. Holes accumulate following the trapping of photo-generated electrons. The charge depletion after the X-ray irradiation follows a slow kinetic due to the removal of electrons from the recombination dynamic by traps. The resulting photoconductive gain effect explains the observed strong electrical response and high sensitivity to X-rays despite the low-Z organic material. The bendability of these flexible detectors is evaluated through measurements of their performance under mechanical stress. A conformable X-ray detection system based on a 2 × 2 pixelated matrix of organic thin film sensors is realized and tested. A good

discrimination between pixels when selectively irradiated by X-rays is assessed and the performance of the device for imaging application is demonstrated.

These results open the way for novel flexible, large area and low voltage ionizing radiation detection systems, capable of providing quantitative and real-time information on the dose rate and on the spatial distribution of impinging X-ray radiation. Thanks to the very low power consumption and room temperature operation, portable applications can be also envisaged.

## Methods

**Devices fabrication.** The X-ray detectors are fabricated on a highly flexible 125 μm thick PET film. First, interdigitated gold electrodes, with a nominal channel length $L = 30 \pm 5$ μm, are thermally evaporated onto the plastic substrate, and their geometry is defined by a standard photolithographic process. Afterwards, the organic active layer is deposited over the whole structure by drop-casting of a 0.5% wt. TIPS-pentacene solution in toluene. During the deposition of the organic solution, the substrate is kept at 60 °C, after the deposition the devices are annealed at 80 °C for 1 h.

Low voltage OTFTs are fabricated using the procedure reported by Cosseddu et al.[41].

**AFM measurements.** AFM measurements are performed on a Park NX10 system using PPP-NCHR tips (Nanosensors) in non-contact mode and applying adaptive scan-rate to slow down scan speed at crystallite borders.

**Electrical measurements.** The electrical photo-response of the detectors is measured by using a Keithley 6,175A electrometer, except for the detector matrix, where a 4-channel picoammeter AH501 is employed to simultaneously record the signal from the four pixels. 2- and 4-probe measurements are performed using a Keithley 2,614 source-meter.

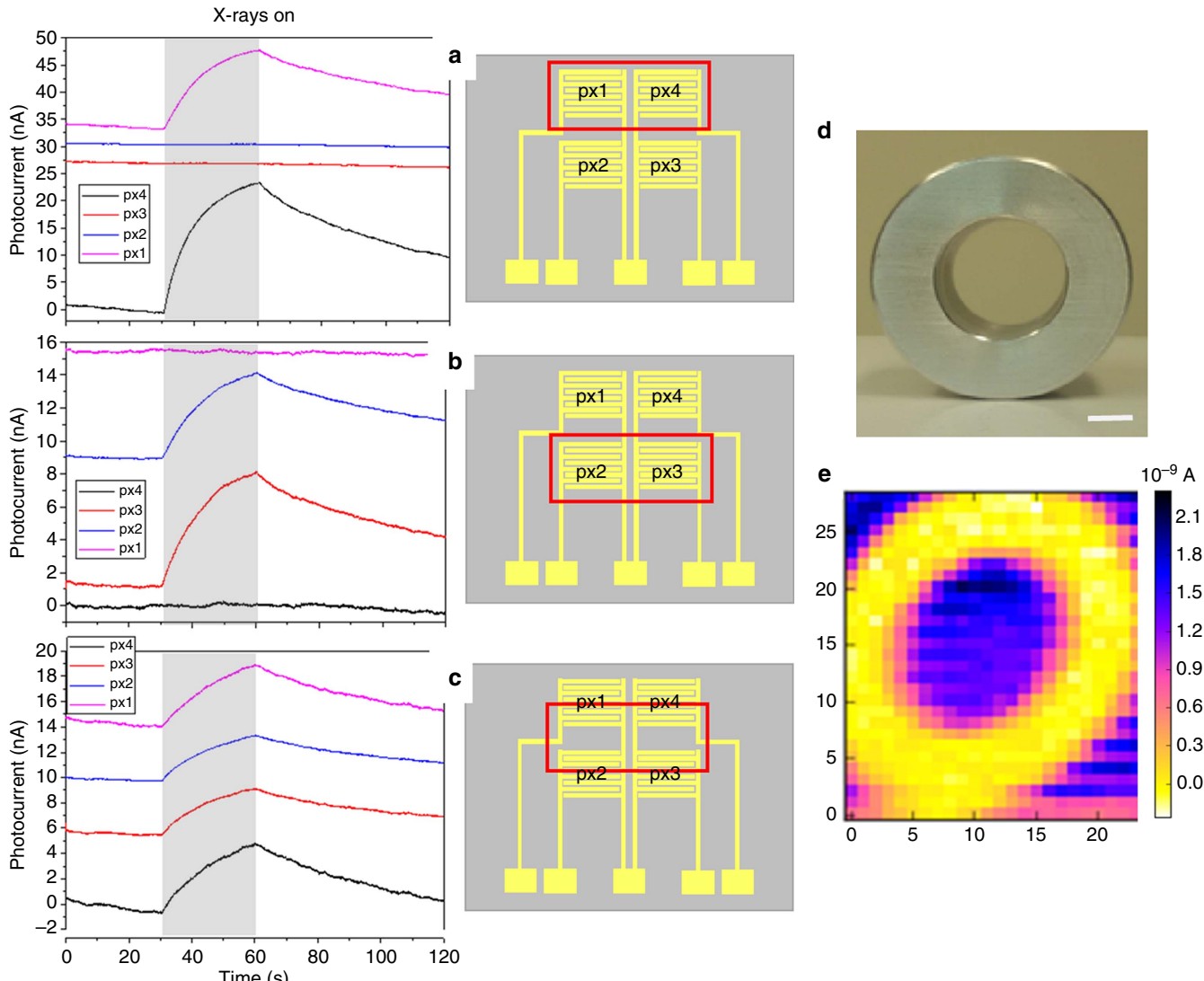

**Figure 5 | 2 × 2 pixel matrix organic detector.** Left: X-ray-induced current signals versus time, recorded by selectively irradiating the pixels of a 2 × 2 detector matrix. The radiation source employed is a monochromatic synchrotron X-ray beam at 17 keV with a dose rate of 28.5 mGy s$^{-1}$. The pixels (with $W = 48$ mm and $L = 30$ μm) were all biased at 0.2 V. Note that the baseline of the four photocurrent signals are shifted in y axis for clarity. Centre: a sketch of the device is reported, with the red box indicating the region of the matrix under irradiation. In particular: in **a** only pixels 1 and 4 are irradiated, in **b** only pixels 2 and 3 are irradiated, in **c** all the pixels are irradiated. Right: photograph (**d**) and corresponding X-ray image by a single pixel device (**e**) of an aluminium annular ring; the scale bar is 5 mm.

**X-ray irradiation.** Two different X-ray beam sources are employed for the characterization of the detectors: (a) a Molybdenum tube X-ray broad spectrum with accelerating voltage of 35 kV and dose rates 2.5—60 mGy s$^{-1}$, and (b) a monochromatic and aligned synchrotron X-ray beam with energy of 17 keV and dose rate in the range 0.05—35 mGy s$^{-1}$. Synchrotron measurements are carried out at ELETTRA—Trieste, in the SYRMEP beamline that is equipped with an ionization chamber for real-time dose rate monitoring. In addition, precision slits and a CCD camera allow to focus the beam and to control the irradiated area of the samples.

**Data availability.** The data that support the findings of this study are available from the corresponding author on request.

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

## Acknowledgements

The authors acknowledge the financial support from the European Community under the FP7-ICT Project 'i-FLEXIS' (2013–2016), Grant Agreement no. 611070. We are grateful to Giuliana Tromba and Diego Dreossi at Elettra synchrotron in Trieste for hardware and software technical support during the experiments, in particular for the acquisition of X-ray image. We also acknowledge Dr Stefano Lai for his precious help in device fabrication.

## Author contributions

P.C., L.B. and A.C conceived and designed the experiments. P.C. and L.B fabricated the devices. P.C. fabricated OTFTs for mobility measurements. L.B., A.C. and P.C. carried out electrical measurements, devices characterization under X-rays and data analysis. A.C. implemented the software for data acquisition and designed the experiments at synchrotron beamline. T.C. performed AFM measurements, developed the kinetic model and carried out numerical fit. A.C., T.C. and L.B. contributed to simulation of experimental curves. L.B. wrote the first draft of the manuscript, with the help of T.C. in the part concerning the kinetic model description. All authors discussed the results and revised the manuscript. B.F. and A.B. coordinated the project.

## Additional information

**Competing financial interests:** The authors declare no competing financial interests.

