## [Peer Review File · Nature Communications]

Reviewers' comments:

Reviewer #1 (Remarks to the Author):

This manuscript reports on direct X-ray photoconversion in flexible, ultra-low voltage organic thin film devices. The manuscript contains some interesting results, however, several questions arise. The authors need to strengthen their scientific explanations for the results they have obtained.

1. In this report, the authors emphasized that the organic thin film based X-ray detector can be operated at ultra-low voltage (0.2 V). However, in my humble opinion, the driving voltage is less important than sensitivity. In other words, the driving voltage should be increased in order to obtain better sensitivity. I recommend the authors to provide the sensitivity of the device at various driving voltages.
2. In the last report, the authors confirmed that an X-ray detector with metal electrodes shows larger photocurrent than all organic sensor. [Fraboni, B. et al. Organic Semiconducting Single Crystals as Next Generation of Low-Cost, Room Temperature Electrical X-ray Detectors. *Adv. Mater.* 24, 2289-2293 (2012)]. In my humble opinion, it seems that the X-ray photocurrent results also may originate from the secondary electrons from the metal electrodes. To strengthen the authors' opinion, metal electrodes of the devices should be shielded from the incoming X-ray beam and photocurrent should be re-measured.
3. Usually, photoconductive gain effect is occurred with prolonged carrier life time. Therefore, carrier mobility should be changed when photoconductive gain effect is occurred because carrier life time is proportional to carrier mobility. However, in this report, carrier mobility exhibits no significant difference under X-ray irradiation. How could it be possible?
4. On page 2, line 9:
..., where the radiation is at first absorbed by an (organic) scintillating material...
The word 'organic' should be changed 'inorganic'.
5. On page 2, line 18:
The authors only mentioned improving stopping power of material in reference to previous works. However, improving carrier mobility is also the subject of previous works. Therefore, the authors should mention all of them.
6. In figure 1C:
The unit of the figure seems to be wrong.
7. On page 3, line 16
The authors should provide statistical data in graph or appropriate form.
8. The author should provide the references of equations. It is difficult to know whether the equations are original works of authors or not.

Reviewer #2 (Remarks to the Author):

This is an interesting piece of work that claims a high level of photoconductive gain for micro-crystalline thin films of TIPS-pentacene. Previous authors have presented similar levels of X-ray induced photocurrent from direct conversion in TIPS-pentacene thick films (eg. refs 18-20) but the novelty of this work is the use of a very thin organic layer and operation at low bias voltage using a coplanar electrode structure.

The key claim in this paper is that a photoconductive gain of $2.6E4$ has been achieved in a typical device. This gain value is deduced from the very long 'effective lifetime' of 29.4s. Referring to the supplementary information, please can the authors provide more information about this calculation, specifically:

- the calculation of the measured charge density ' ρ_{-x} ' from ' ΔI_{-PG} '. Last equation on p1. Is this equation valid for a steady-state photocurrent measured over a 60s period

- the calculation of ' τ_{-r} '. 1st equation at top of p2. Please explain the 'fitting parameters' used in this equation, eg. ' α ', ' γ ' and ' ρ_{-0} ', and also with reference to equation S2. What are the uncertainties of these fitted parameters, and hence what is the uncertainty of the ' τ_{-r} ' value of 29.4s?

Reviewers' comments:

Reviewer #1 (Remarks to the Author):

In the revised manuscript, authors improved their scientific contents and opinions. However, the evidences and scientific explanations to support their claim are still insufficient and inappropriate.

1. The authors insisted that high bias voltage can affect the device stability but there is no supporting data for their device. Indeed, in the revised manuscript, the photocurrent shows increasing tendency until 20 V. Also, this devices doesn't show reduced photocurrent in the regime of space charge limited current (SCLC). Therefore, with the current experimental data, it is still difficult to understand why this device should be operated at ultra-low voltage. In my opinion, to emphasize using ultra-low voltage for X-ray detector, appropriated explanation should be inserted in introduction part and the X-ray sensitivity of the device at ultra-low voltage should be compared with other organic X-ray detectors.

2. To confirm the effect of electrodes under X-ray irradiation, the authors made dummy sample. However, it is not appropriate to compare the device with dummy sample due to the dummy sample structure. Unlike the original device structure, in dummy sample structure, the potential between two electrodes has little effect to current of device due to the absence of electrical channel. The electrical path between two electrodes should be exist to confirm the secondary electrons produced by the X-ray irradiation. Therefore, to confirm the electrodes effect, X-ray induced photocurrent of the devices with shielded electrodes should be measured. If this is not possible due to the device structure, find another ways such as deposition electrodes onto the TIPS pentacene layer or using another shape of electrode or shielding the electrodes from bottom of the substrate.

3. For figure 2b, the authors insisted that the X-ray exposure has little impact on interfacial properties between TIPS-pentacene and electrode. According to the manuscript, relative variation of the conductance is 40% under X-ray exposure for both the 4- and 2-probe measurement. For this device structure, there are two resistance, one is bulk resistance (R_b) and another is contact resistance (R_c). In case of 4-probe measurement, calculated resistance shows bulk resistance without contact resistance. In case of 2-probe measurement, calculated resistance shows the combination of bulk and contact resistance. If an X-ray exposure has little impact on interfacial properties, contact resistance doesn't change under the X-ray exposure. Then, the relative variation of the conductance under X-ray exposure (R_{bx} and R_{cx}) for both the 4- and 2-probe measurement should be different.

For 2-probe measurement

$$R_{bx}/R_b \times 100\% = 40\%$$

For 4-probe measurement

$$(R_{bx} + R_{cx}) / (R_b + R_c) \times 100\% \neq 40\% \text{ (If } R_{cx} = R_c)$$

Therefore, the results of 4- and 2-probe measurements doesn't mean that X-ray has no impact on interfacial properties.

4. In the revised manuscript, the author said that organic materials have been mostly employed in indirect radiation detection system. However, there are many kinds of inorganic scintillators such as NaI, CsI, and BGO etc. They are already used as commercial X-ray detectors. Therefore, the expression should be changed.

REVIEWERS' COMMENTS:

Reviewer #1 (Remarks to the Author):

This manuscript proposed a photoconductive gain model to interpret the unexpected outcome photocurrent result which is larger than estimated photocurrents based on X-ray photon absorption of low-Z organic material and photo conversion effect. In a way the experimental results seem to be appropriate for supporting the author's claim. However it is still not enough to publish in this journal. The given data are inconsistent, inaccurate, or misleading. In my opinion, the whole manuscript should be re-organized and the figures should be improved. For example, in Fig. 5a-c, the base currents (dark currents) of each 4-pixels show different values for the different measurements. If the authors intended to show 4 data in one graph, the graph should be drawn differently. The current form show that the 4 pixels have similar photocurrent values with a large difference in dark current. Inappropriate form of expression lower the claim and experimental results reliability. Also, it has still miss descriptions in this manuscript (for example, Fig.4b y-axis unit).

Reviewer #2 (Remarks to the Author):

As requested, I have looked at Referee #1's comments on the first revision, and the authors responses to these latest comments:

1. The referee was again questioning the validity of claiming that there is a benefit in operating this device at low voltage. I can understand this concern since the original paper did not give a good justification for low voltage operation, other than reducing bulk current and signal noise. However the new supplementary fig 3 gives good evidence of higher bulk current at $V > 2v$, which when compared to the linear increase in photocurrent in the same voltage range, causes reduced S/N at $V > 2v$. In addition, Fig 2 seems to also confirm that the device has significantly better long term stability at low bias voltage, compared to higher voltages. Taken together, these are reasonable arguments to support operating the device at $V < 2v$.
2. The authors have taken new data to investigate the possible enhancement to the X-ray induced photocurrent due to photoelectron production from the metal electrodes. Their new graph shows that the X-ray photocurrent is enhanced by $\sim 100\%$ using gold electrodes compared to PEDOT electrodes, which is an expected outcome. Does this result undermine their paper? Perhaps not, although I suggest they should include this new figure into the supplementary information. The referee asked for more evidence about this effect, and this has been provided.
3. A comment about 2-probe vs 4-probe measurements. Both parties seem to have agreed with each other, and a minor modification has been made to the text.
4. This is a trivial misunderstanding about the emphasis of the sentence, which has been resolved.

My view is that the authors have met the requirements of the referee. Only points 1 and 2 were significant issues and they have answered these questions reasonably well. In my opinion the major importance of this paper remains the fact that the authors report high photoconductive gains and long carrier lifetimes. I find these results surprising and impressive, so I think the paper should be published to let the community see the data.

Dear Senior Editor,
Our responses to Reviewers are highlighted below each comment.

Reviewers' comments:

Responses to Reviewer#1 (Remarks to the Author):

This manuscript reports on direct X-ray photoconversion in flexible, ultra-low voltage organic thin film devices. The manuscript contains some interesting results, however, several questions arise. The authors need to strengthen their scientific explanations for the results they have obtained.

1. In this report, the authors emphasized that the organic thin film based X-ray detector can be operated at ultra-low voltage (0.2 V). However, in my humble opinion, the driving voltage is less important than sensitivity. In other words, the driving voltage should be increased in order to obtain better sensitivity. I recommend the authors to provide the sensitivity of the device at various driving voltages.

1. We agree with the reviewer that sensitivity is a crucial parameter in X-ray detectors. We measured the signal amplitude for different bias voltages and we now provided the plot of photocurrent vs. bias voltage, labeled as Fig.1e in the manuscript, showing an almost linearly increasing behavior in good agreement with the photoconductive gain model. Indeed, higher driving voltages would lead to an increase in sensitivity.

However, three major issues prevent to exploit high bias voltages in organic photoconductive detectors:

1. Organic materials significantly suffer from bias stress [IEEE Trans. On Elec. Devices 57, 5, 2010], even at a few bias volts [U. Zschieschang et al., Appl. Phys. A 95, 139–145, 2009]. As a consequence, the performance of our organic devices at higher voltages (>1V) may be limited in terms of reproducibility and reliability. Since the measurements needed to estimate the detector sensitivity are quite time consuming and require continuous biasing of the device, providing reliable sensitivity values and good detector performance at high voltages is not trivial. Furthermore, the thermal stability of the system would not be guaranteed anymore at high biases. In the main text a statement explaining this point has been added (page3 line 24).
2. The IV (current vs. voltage) characteristics in dark of our devices show a supralinear behavior for $V > 2V$, due to the onset of a space charge limited current regime (SCLC) within the organic material. In the Supplementary Information section we included a new figure (Supplementary Fig.3) that reports the typical IV curve of our devices in dark showing the onset of the SCLC regime. In the text we added a sentence referring to it (page 4 line 2).
3. Last but not least, although sensitivity is crucially important, the main goal of this technology is to provide devices able to work at very low voltages, in order to be portable, light-weight, low cost, even disposable. Therefore, under the application point of view, finding a suitable trade-off between these requirements and sensitivity is mandatory.

2. In the last report, the authors confirmed that an X-ray detector with metal electrodes shows larger photocurrent than all organic sensor. [Fraboni, B. et al. Organic Semiconducting Single Crystals as Next Generation of Low-Cost, Room Temperature Electrical X-ray Detectors. Adv. Mater. 24, 2289-2293 (2012)]. In my humble opinion, it seems that the X-ray photocurrent results also may origin from the secondary electrons from the metal electrodes. To strengthen the authors' opinion, metal electrodes of the devices should be shielded from the incoming X-ray beam and photocurrent should be re-measured.

2. We agree with the reviewer, the contribution of the metal electrodes to the X-ray photocurrent has to be considered. Our devices have a bottom contact configuration (see Fig.1a in the manuscript), i.e. the gold electrodes lie underneath the organic thin film, therefore shielding electrodes is impractical in this geometry. However, to address the reviewer's concern, we provide the following additional measurements and considerations:

1. We carried out X-rays induced photocurrent measurements under identical experimental conditions on fully operating devices and on "dummy" devices, fabricated onto the same substrate and electrodes but without any organic layer. We recorded a non-zero signal from the "dummy" devices probably due to air ionization effect induced by secondary electrons emitted from the metal electrodes. We reckon that this is the most direct measurement to assess the contribution to the X-ray photocurrent from the secondary electrons. We have added a figure in supplementary information (Supplementary Fig.2) showing the photocurrent signal of a dummy device upon exposure to a 25 mGy/s synchrotron radiation at 17 keV, recorded for different bias voltages: 0.2V, 0.5V, 1V, 5V, 10V, 20V. The data clearly show that the electrodes contribution to the photocurrent is about one order of magnitude lower than that recorded with the organic device (Fig. 1d), even at a higher intensity of radiation and with a much higher bias voltage ($\Delta I \approx 0.3 \text{ nA}$ at 20V of bias voltage employing a 25 mGy/s dose rate X-ray beam at 17keV). We have also added a sentence in the main text (page3 line21) referring to the Figure in Supplementary Information.
2. Further, secondary electrons could indirectly increase the X-ray induced signal by what is called "dose enhancement effect", i.e. secondary electrons ionize organic molecules in the close proximity (few tens of nanometers) of the metal electrodes. From the 4-probes and 2-probe conductance measurements carried out during switching on and off the X-ray beam, reported in Fig. 2b, we can infer that the contribution of the electrodes or electrode interfaces is not dominant and the main variation of the conductance of the active layer under X-ray irradiation happens in the bulk of the semiconductor.

3. Usually, photoconductive gain effect is occurred with prolonged carrier life time. Therefore, carrier mobility should be changed when photoconductive gain effect is occurred because carrier life time is proportional to carrier mobility. However, in this report, carrier mobility exhibits no significant difference under X-ray irradiation. How could it be possible?

3. The comment of the referee regarding the relation between carrier lifetime and mobility is correct and his/her question is very useful. To answer this point we emphasize that, in the described organic X-ray detector, one has to distinguish transport of holes and electrons. In fact, in organic p-type conductors such as TIPS pentacene macroscopic charge transport occurs almost exclusively via holes as discussed in the manuscript (page5 line17). Any mobility characterization by means of transfer characteristics therefore assesses hole mobility. The results we present in this manuscript thus show that hole mobility does not vary under X-ray irradiation. Now, the observed photoconductive gain effect relies on the prolonged life time of electron trap states. Of course, the lifetime variation in these states will affect electron carrier mobility. However, as electron mobility is orders of magnitude lower than hole mobility in TIPS pentacene, these variations are not measurable. On the opposite, the holes transport prevails and the photocurrent under X-ray irradiation results from an increased hole carrier density counterbalancing the increased amount of trapped electron charges (due to long life time of electron traps).

To emphasize the crucial distinction between electron and hole transport in TIPS pentacene we now refer separately to hole and electron mobility (page5 lines4-5 in the main text and in Supplementary

Information in the caption of Fig.4). In addition, we enforced arguments in paragraph at page 5 to avoid possible misunderstandings.

4. On page 2, line 9:

..., where the radiation is at first absorbed by an (organic) scintillating material...

The word 'organic' should be changed 'inorganic'.

4. Indeed, in the sentence the reviewer mentions, we meant to refer also to the employment of organic materials as scintillators (or as active layer of photodetectors) in indirect radiation detector systems. Traditionally, organic materials were suggested as scintillators, in the form of molecules dispersed in either liquids [H. Kallman, Phys. Rev. 1950, 78, 621] or in plastic matrices [G. Bertrand et al., Chem. Eur. J. 2014, 20, 15660], or as polycrystalline material [S. V. Budakovskiy et al., Mol. Cryst. Liq. Cryst. Sci. Technol., Sect. A 1998, 324, 145]. In addition, some studies reported on solution grown organic single crystals as scintillators for radiation detectors, in particular evaluating their gamma/neutron pulse shape discrimination properties [G. Hull et al., IEEE Trans. Nucl. Sci. 2009, 56, 899]. Moreover, also inorganic scintillating particles can be employed coupled to an organic photodetector in indirect ionizing radiation detection system as recently reported [Bücheler, P. et al. Nat. Photonics 9, 843–848 (2015)]. The sentence on page 2 line 7 has been modified in order to clarify this point.

5. On page 2, line 18:

The authors only mentioned improving stopping power of material in reference to previous works. However, improving carrier mobility is also the subject of previous works. Therefore, the authors should mention all of them.

5. The text has been slightly changed to address reviewer's concern. (page 2 line 19)

6. In figure 1C:

The unit of the figure seems to be wrong.

6. The reviewer is right. The unit of the x-axis in the plot has been changed in accordance to reviewer's comment.

7. On page 3, line 16

The authors should provide statistical data in graph or appropriate form.

7. In accordance to the reviewer's suggestion, we included in Supplementary Information section a new figure labeled Supplementary Fig.1, reporting the sensitivity values on a histogram. The values are distributed with a statistical variation <15% within the average.

8. The author should provide the references of equations. It is difficult to know whether the equations are original works of authors or not.

8. We agree with the reviewer. Indeed, a reference to a textbook chapter on photoconductive gain containing the basic formulas employed in our manuscript was missing. All of the basic equations on photoconductive gain can be found in the textbook *Physics of optoelectronic devices* (Wiley & Sons, Inc.) of Shun Lien Chuang, chapter 14.1 "Photoconductors". In the revised manuscript, we refer to this book on several occasions. In other parts, equations are original works. In that case, we refer to the Supplementary Information section where further information on the derivation of the equation can be found.

Responses to Reviewer#2 (Remarks to the Author):

This is an interesting piece of work that claims a high level of photoconductive gain for micro-crystalline thin films of TIPS-pentacene. Previous authors have presented similar levels of X-ray

induced photocurrent from direct conversion in TIPS-pentacene thick films (eg. refs 18-20) but the novelty of this work is the use of a very thin organic layer and operation at low bias voltage using a coplanar electrode structure.

The key claim in this paper is that a photoconductive gain of 2.6×10^4 has been achieved in a typical device. This gain value is deduced from the very long 'effective lifetime' of 29.4s. Referring to the supplementary information, please can the authors provide more information about this calculation,

Below we provide detailed answers to his questions on the calculation of the gain based on the effective lifetime of carriers. In this context we also would like to emphasize that a lower limit for the gain G can also be estimated directly following equation (1): $G = \Delta I_{PG} / I_{CC}$, with the amount of charge created by the ionizing radiation per time in the organic semiconductor ($I_{CC} < I_{CC,max} = 2 \text{ pA}$) estimated following Lambert-Beer law to provide the amount of absorbed X-ray photons $\Phi = \frac{DA}{c_m E_{ph}} (1 - e^{-h/\lambda})$ and energy conservation to compute the number of electron/hole pairs created per absorbed photon: $n = E_{ph} / E_g$ (see page 4 in the manuscript). Following this calculation we reach $G > 3 \text{ nA} / 2 \text{ pA} = 1500$. The actual value for G is higher due to a reduced quantum efficiency in photon to excitonic charge conversion ($\sim 10\%$).

specifically:

- the calculation of the measured charge density 'rho-x' from 'delta-I-PG'. Last equation on p1. Is this equation valid for a steady-state photocurrent measured over a 60s period

-Yes, the equation is widely applied to describe the response of uniform photoconductors with mobility independent on carrier density and only contributions due to drift components (no spatial gradients in ρ_x present). It is introduced for example in Ref.25 [Shun Lien Chuang, *Physics of optoelectronics devices*. (Wiley & Sons, Inc.)], chapter 14. In our experiments these conditions are fulfilled and the ionizing radiation is spatially homogeneous on the length-scale of the channel. Further, in the calculation of ρ_x we used the general expression of ΔI_{PG} described by equation (2), substituting the experimental value of the photocurrent signal amplitude $\Delta I_{PG} \approx 3 \text{ nA}$ recorded after 60 s exposure of the device to a 19.3 mGy/s dose rate X-ray beam at 17keV, i.e. the steady-state photocurrent. The experimental curve we refer to in such a calculation is that reported in Fig.1d, where photocurrent value of about 3 nA, recorded after 60s of exposure in three subsequent on/off switching cycles of the X-rays is shown. In the Supplementary Information section (page2 line13) we modified the text explicitly referring to the experimental value of the steady-state photocurrent ΔI_{PG} used in the calculation.

- the calculation of 'tau-r'. 1st equation at top of p2. Please explain the 'fitting parameters' used in this equation, eg. 'alpha', 'gamma' and 'rho-0', and also with reference to equation S2. What are the uncertainties of these fitted parameters, and hence what is the uncertainty of the 'tau-r' value of 29.4s?

- The reviewer refers to the equation (4):

$$\tau_r = \frac{\alpha}{\gamma} \left[\alpha \ln \left(\frac{\rho_0}{\rho_x} \right) \right]^{\frac{1-\gamma}{\gamma}}$$

This equation originates from the finding that the photocurrent, and hence the excess charge density ρ_x , decay follow a stretched exponential during relaxation in dark (X-ray irradiation turned off), as detailed in the manuscript on page 7 and 8:

$$\rho_x = \rho_0 e^{-t^\gamma/\alpha}$$

From this equation, the physical interpretation of the fitting parameters becomes clear:

ρ_0 is an initial reference charge density; α is a characteristic time scale of the relaxation of charge carriers. In particular, when starting with an initial charge density of $\rho_X(t = 0) = \rho_0$, then $\rho_X(t = \alpha^{1/\gamma}) = \rho_0/e$.

Finally, γ is a parameter describing the width of the distribution ρ_i of time-scales α_i involved in the relaxation dynamics: for the case of $\gamma = 1$ only one time-scale is present and the dark relaxation kinetics follows an exponential behavior. Instead, for $\gamma < 1$ the distribution gets broader. For the sample analyzed in the manuscript we observe $\gamma = 0.61$. Therefore, initially a fast relaxation occurs due to holes recombining with electrons escaping from shallower traps. As the charge density reduces, the escape process slows down, as holes have to recombine with electrons escaping from deeper traps. The distribution of time-scales can be calculated by the inverse Laplace transform of the stretched exponential relaxation kinetics. Some characteristic distributions are shown in the graphs below:

Clearly, a distribution of time-scales results from disorder in the organic semiconductor. As it is a micro-crystalline film, we expect a possible role of grain boundaries or surface states in the trapping mechanism.

In order to transform the time scales to activation energies, one could exploit the Arrhenius equation:

$$\alpha_i = \frac{1}{\nu} \exp\left(\frac{E_{A,i}}{kT}\right)$$

However, here the attempt frequency ν is unknown, therefore we don't further pursue these arguments.

The analysis based on stretched exponential behavior was originally conceived for the analysis of photoconductive transients in amorphous semiconductors. More details can be found in the corresponding literature: Jiajun Luo et al., J. Apl. Phys. 113, 153709 (2013); D. Redfield and R. H. Bube, *Photoinduced Defects in semiconductors* (Cambridge University Press, 1996). In the manuscript we added further explanatory sentences about the physical interpretation of the fitting parameters and we included also a further paragraph in the Supporting Information section (page1) showing the above figure (Supplementary Fig.6).

Next, the referee refers to equation S2, derived in the Supplementary Information section:

$$D = c \Delta I_{PG} \left[\ln\left(\frac{I_0}{\Delta I_{PG}}\right) \right]^{\frac{\gamma-1}{\gamma}}$$

This equation shows a non-linear dependence between X-ray dose rate D and saturation photoresponse ΔI_{PG} . The origin of the non-linearity is the above discussed distribution of carriers relaxation timescales characterized by the parameters ρ_0, α, γ . The former two enter indirectly as the pre-factor $c = \frac{\tau_t \gamma}{S_{CC} \alpha^{1/\gamma}}$ and the reference current $I_0 = Wh\rho_0\mu E$, as described in the Supplementary

Information section (page 4 lines1-2). At low dose-rates holes recombination happens with deep electron traps and thus τ_r becomes larger, resulting in higher gain and sensitivity. In contrast, at high dose rates and large ρ_X , also shallow trap states are involved thus overall relaxation happens faster and gain is lower.

Finally, we address uncertainties in parameters ρ_0, α, γ . Uncertainties are expressed as standard variations obtained by fitting numerically photocurrent transients of different devices. Stable three parameters fits are obtained by fitting the stretched exponential decay of dark relaxation measured for at least 60 s. We obtain $\alpha=(8.0 \pm 1.3)$ s, $\gamma=0.60 \pm 0.07$, $\rho_0=(3.7 \pm 0.4) \times 10^{-5}$ C cm⁻³. These uncertainties have been introduced in the manuscript (page7 line24).

The uncertainty of τ_r is dose rate dependent as it depends on the charge density of photogenerated carriers: $\tau_r(\rho_X)$. For $D=19$ mGy/s we obtain, considering averages over different devices: $\tau_r(\rho_X) = (30 \pm 4)$ s.

Best Regards,

Dr. Laura Basiricò in behalf of all authors

Dear Editor,

Our responses to the Reviewer are highlighted below each comment.

Reviewers' comments:

Reviewer #1 (Remarks to the Author):

In the revised manuscript, authors improved their scientific contents and opinions. However, the evidences and scientific explanations to support their claim are still insufficient and inappropriate.

1. The authors insisted that high bias voltage can affect the device stability but there is no supporting data for their device. Indeed, in the revised manuscript, the photocurrent shows increasing tendency until 20 V. Also, this devices doesn't show reduced photocurrent in the regime of space charge limited current (SCLC). Therefore, with the current experimental data, it is still difficult to understand why this device should be operated at ultra-low voltage. In my opinion, to emphasize using ultra-low voltage for X-ray detector, appropriated explanation should be inserted in introduction part and the X-ray sensitivity of the device at ultra-low voltage should be compared with other organic X-ray detectors.

1. Authors agree with the reviewer's comment. In fact this point deserves a better explanation. It is well known that bias stress decreases the current in the organic semiconductor due to charge trapping, moreover charge trapping increases its rate as the bias increases. This effect is clear from the measurements we have now reported in the Supplementary Information section, where we added a figure, labeled Supplementary Fig.2 reporting the normalized current decreasing with a fixed applied voltage of 1V (black squares), of 10V (red circles) and of 20V (green triangles) in function of time. The plot clearly shows that the current at 1V bias shows only minor drift and it remains stable for 20min of continuous biasing. Current curves obtained at higher bias voltages show instability and the current drops quickly decreasing of about 50% at 20V. We added in the main text the reference to the new Supplementary figure (page4 lines1-2).

The onset of the space charge effect at $V > 2V$ leads to a supralinear increase of the dark current (Supplementary Fig.3), while the device photocurrent increases linearly at higher voltages (Fig. 1e), as the Reviewer correctly observed. The important issue here is that when operating at high voltages, the net result of such different trends is the strong decrease of the detector signal-to-noise ratio, i.e. the ratio between the X-rays induced photocurrent and the dark current. This is obviously a detrimental effect on the overall detector performance. In the revised manuscript, we explicitly explain this issue (page4, line2).

In the last sentence of page 9, we provided specific references for the comparison of the X-ray sensitivity of our device and other organic direct X-ray detectors reported in literature. The major and relevant issue here is that, to our knowledge, no other low-voltage organic direct X-ray detectors have been reported so far in literature. None of the other reported detectors operate at low bias but they operate at several tens of Volts. Therefore, we could not compare our sensitivity values with any other reported data, for the same low-voltage condition. We added a sentence to the text to clarify this point (page10, line1).

In the introduction a sentence to emphasize the use of ultra-low voltage for X-ray detectors has been inserted following the Reviewer's suggestion (page 3 line 5).

2. To confirm the effect of electrodes under X-ray irradiation, the authors made dummy sample. However, it is not appropriate to compare the device with dummy sample due to the dummy sample structure. Unlike the original device structure, in dummy sample structure,

the potential between two electrodes has little effect to current of device due to the absence of electrical channel. The electrical path between two electrodes should be exist to confirm the secondary electrons produced by the X-ray irradiation. Therefore, to confirm the electrodes effect, X-ray induced photocurrent of the devices with shielded electrodes should be measured. If this is not possible due to the device structure, find another ways such as deposition electrodes onto the TIPS pentacene layer or using another shape of electrode or shielding the electrodes from bottom of the substrate.

2. The Reviewer made a comment on this issue in his first revision, referring to our previous work [Fraboni, B. et al. Organic Semiconducting Single Crystals as Next Generation of Low-Cost, Room Temperature Electrical X-ray Detectors. *Adv. Mater.* 24, 2289-2293 (2012)], where the role of metal electrodes to the collected electrical signal was considered as shown by the figure reported below:

The signal amplitude of an organic device with metal electrodes was compared with that of the device with shielded electrodes and with that of a full-organic detector (with PEDOT:PSS electrodes). The figure shows that, even though a non-negligible contribution of electrodes is observed, accounting for secondary electrons produced by the X-ray irradiation, the signal from the full-organic device is comparable to that obtained from devices with shielded metallic electrodes, confirming that organic single crystals exposed to X-rays can directly convert the incoming radiation into an electrical signal with no need for additional high-Z components in the device. Therefore, in order to further and better address the reviewer's request, we repeated the same experiment comparing a TIPS-pentacene thin film device with metal electrodes (gold) with a full-organic one (PEDOT:PSS electrodes). Both the devices have been biased with 1 V and have the same electrode geometry. The results, reported below, are in accordance with those previously reported for organic single crystals: the occurrence of the direct photoconversion effect in the organic crystals.

Moreover, the reviewer's suggestions on possible device shielding processes, beside being impractical due to the present device structure, cannot be taken as viable and effective ways to distinguish the role of secondary electrons emitted by the electrodes to that of the bulk material. In fact, shielding of the electrodes from the bottom of the substrate or using the top-contact configuration with shielded electrodes would prevent the radiation to reach the organic material covering the electrodes, and thus it would exclude the signal coming from those areas of the active layer, not only the secondary electrons emitted from the electrodes.

3. For figure 2b, the authors insisted that the X-ray exposure has little impact on interfacial properties between TIPS-pentacene and electrode. According to the manuscript, relative variation of the conductance is 40% under X-ray exposure for both the 4- and 2-probe measurement. For this device structure, there are two resistance, one is bulk resistance (R_b) and another is contact resistance (R_c). In case of 4-probe measurement, calculated resistance shows bulk resistance without contact resistance. In case of 2-probe measurement, calculated resistance shows the combination of bulk and contact resistance. If an X-ray exposure has little impact on interfacial properties, contact resistance doesn't change under the X-ray exposure. Then, the relative variation of the conductance under X-ray exposure (R_{bx} and R_{cx}) for both the 4- and 2-probe measurement should be different.

For 2-probe measurement

$$R_{bx}/R_b \times 100\% = 40\%$$

For 4-probe measurement

$$(R_{bx} + R_{cx}) / (R_b + R_c) \times 100\% \neq 40\% \text{ (If } R_{cx} = R_c \text{)}$$

Therefore, the results of 4- and 2-probe measurements doesn't mean that X-ray has no impact on interfacial properties.

3. The referee is right, the interfacial properties and contact resistance are affected by X-rays. The effects are of the same size for bulk and contact resistance variation (40%), showing that both can be explained by the same amount of modulation in carrier density. The experiment and the related argument had a different objective: it was to demonstrate that interfacial effects do not prevail, i.e. the bulk contribution is not zero as would be the case if the carrier density increase would be only caused by dose enhancement effects close to electrodes (i.e. by effects induced by the presence of metal electrodes).

4. In the revised manuscript, the author said that organic materials have been mostly employed in indirect radiation detection system. However, there are many kinds of inorganic

scintillators such as NaI, CsI, and BGO etc. They are already used as commercial X-ray detectors. Therefore, the expression should be changed.

4. In the sentence the Reviewer refers to, our intention was to explain the possible and reported employment of organic materials in ionizing detection systems, which has been more common in indirect systems than in direct ones so far. We did not mean that organic scintillators are more used than inorganic scintillators in indirect detectors. We modified the text in the introduction (page 1 line 7) to avoid any misleading interpretation.

Dear Editor,

our responses to the issues raised by the reviewers are highlighted below:

Reviewer #1:

This manuscript proposed a photoconductive gain model to interpret the unexpected outcome photocurrent result which is larger than estimated photocurrents based on X-ray photon absorption of low-Z organic material and photo conversion effect. In a way the experimental results seem to be appropriate for supporting the author's claim. However it is still not enough to publish in this journal. The given data are inconsistent, inaccurate, or misleading. In my opinion, the whole manuscript should be re-organized and the figures should be improved. For example, in Fig. 5a-c, the base currents (dark currents) of each 4-pixels show different values for the different measurements. If the authors intended to show 4 data in one graph, the graph should be drawn differently. The current form show that the 4 pixels have similar photocurrent values with a large difference in dark current. Inappropriate form of expression lower the claim and experimental results reliability. Also, it has still miss descriptions in this manuscript (for example, Fig.4b y-axis unit).

Indeed, in Fig.5a-c, the four signals of the pixels reported are all referred to the X-rays induced photocurrent, i.e. the dark current have been subtracted, and the baseline of each plot has been shifted in y-axis for clarity. Therefore, the different base current values for the different measurements are simply due to different y-scales. We added a sentence in the main text (page11 line5) and in the figure caption, to clarify the figure's description to the reader and thus to avoid any misleading interpretation.

Regarding the issue about Fig.4b y-axis unit missing description, we honestly do not understand the referee's concern.

Reviewer #2:

As requested, I have looked at Referee #1's comments on the first revision, and the authors responses to these latest comments:

1. The referee was again questioning the validity of claiming that there is a benefit in operating this device at low voltage. I can understand this concern since the original paper did not give a good justification for low voltage operation, other than reducing bulk current and signal noise. However the new supplementary fig 3 gives good evidence of higher bulk current at $V > 2v$, which when compared to the linear increase in photocurrent in the same voltage range, causes reduced S/N at $V > 2v$. In addition, Fig 2 seems to also confirm that the device has significantly better long term stability at low bias voltage, compared to higher voltages. Taken together, these are reasonable arguments to support operating the device at $V < 2v$.

2. The authors have taken new data to investigate the possible enhancement to the X-ray induced photocurrent due to photoelectron production from the metal electrodes. Their new graph shows that the X-ray photocurrent is enhanced by $\sim 100\%$ using gold electrodes

compared to PEDOT electrodes, which is an expected outcome. Does this result undermine their paper? Perhaps not, although I suggest they should include this new figure into the supplementary information. The referee asked for more evidence about this effect, and this has been provided.

Following the referee suggestion, we included the figure in the supplementary information section (labeled as Supplementary Fig.2) and added a sentence in the main text (page3 line23) referring to it and to the contribution of the electrodes to the recorded X-rays photocurrent.

3. A comment about 2-probe vs 4-probe measurements. Both parties seem to have agreed with each other, and a minor modification has been made to the text.

4. This is a trivial misunderstanding about the emphasis of the sentence, which has been resolved.

My view is that the authors have met the requirements of the referee. Only points 1 and 2 were significant issues and they have answered these questions reasonably well. In my opinion the major importance of this paper remains the fact that the authors report high photoconductive gains and long carrier lifetimes. I find these results surprising and impressive, so I think the paper should be published to let the community see the data.